# Red2Flpe-SCON: a versatile, multicolor strategy for generating mosaic conditional knockout mice

Szu-Hsien Sam Wu[1,2], Somi Kim [3], Heetak Lee [4], Ji-Hyun Lee[4], So-Yeon Park[4], Réka Bakonyi[1], Isaree Teriyapirom[1,2], Natalia Hallay[1], Sandra Pilat-Carotta[1], Hans-Christian Theussl[5], Jihoon Kim[4,6], Joo-Hyeon Lee [7,8], Benjamin D. Simons [7,9,10], Jong Kyoung Kim [3], Gabriele Colozza [1] ✉ & Bon-Kyoung Koo [3,4] ✉

Image-based lineage tracing enables tissue turnover kinetics and lineage potentials of different adult cell populations to be investigated. Previously, we reported a genetic mouse model system, *Red2Onco*, which ectopically expressed mutated oncogenes together with red fluorescent proteins (RFP). This system enabled the expansion kinetics and neighboring effects of onco-genic clones to be dissected. We now report Red2Flpe-SCON: a mosaic knockout system that uses multicolor reporters to label both mutant and wild-type cells. We develop the *Red2Flpe* mouse line for red clone-specific Flpe expression, as well as the FRT-based SCON (Short Conditional IntrON) method to facilitate tunable conditional mosaic knockouts in mice. We use the Red2Flpe-SCON method to study Sox2 mutant clonal analysis in the esopha-geal epithelium of adult mice which reveal that the stem cell gene, Sox2, is less essential for adult stem cell maintenance itself, but rather for stem cell pro-liferation and differentiation.

Adult tissue homeostasis is regulated by resident stem cells, which divide at various frequencies to either self-renew or differentiate. Competition between stem cells for niche spaces occurs in many organs throughout life[1-3]. With advances in sequencing technologies and model systems, it has become clear that cell competition plays an important role in regulating tissue turnover and cancer development[4-6]. While prospective or retrospective computational-based lineage reconstruction enables high-throughput analyses of clonal information from tissues, the precise spatial distribution and morphology of the clones can only be revealed using image-based lineage tracing[7].

Multicolor reporter systems have been crucial to understanding the turnover kinetics that occur during tissue homeostasis[8-12]. Com-bining fluorescent reporters with conditional gene knockout (cKO) alleles enables gene knockouts and lineage tracing to be carried out simultaneously[13,14], while utilizing two recombinases in a single mouse

[1]Institute of Molecular Biotechnology of the Austrian Academy of Sciences (IMBA), Vienna BioCenter (VBC), Dr. Bohr-Gasse 3, 1030 Vienna, Austria. [2]Vienna BioCenter PhD Program, Doctoral School of the University of Vienna and Medical University of Vienna, Vienna, Austria. [3]Department of Life Sciences, Pohang University of Science and Technology (POSTECH), Pohang 37673, Republic of Korea. [4]Center for Genome Engineering, Institute for Basic Science, Expo-ro 55, Yuseong-gu, Daejeon 34126, Republic of Korea. [5]IMP/IMBA Transgenic Service, Institute of Molecular Pathology (IMP), Vienna, Austria. [6]Department of Medical and Biological Sciences, The Catholic University of Korea, Bucheon 14662, South Korea. [7]Wellcome–MRC Cambridge Stem Cell Institute, Jeffrey Cheah Biomedical Centre, University of Cambridge, Cambridge, UK. [8]Department of Physiology, Development and Neuroscience, University of Cambridge, Cambridge, UK. [9]The Wellcome Trust/Cancer Research UK Gurdon Institute, University of Cambridge, Tennis Court Road, Cambridge CB2 1QN, UK. [10]Department of Applied Mathematics and Theoretical Physics, Centre for Mathematical Sciences, Wilberforce Road, Cambridge CB3 0WA, UK. ✉e-mail: gabriele.colozza@imba.oeaw.ac.at; koobk@ibs.re.kr

provides further control[15,16]. However, such an approach carries the risk of genotype-phenotype mismatches. Although approaches that use in utero or in vivo electroporation[17] and transduction[18] have been developed – circumventing the need for additional mouse lines – they are fundamentally limited, as they target cell populations in an unspecific manner.

To achieve precise mosaic genetic labeling and tracing, several systems have been developed including, mosaic analysis with double markers (MADM)[19,20], IfgMosaics[21] and Red2Onco[22]. These systems provide a near-definitive genotype-phenotype correlation. Moreover, in these mosaic systems, both mutant and wild-type clones are fluorescently labeled; this enables direct comparisons between lineage kinetics and any potential non-cell-autonomous effects (Supplementary Table 1). Although mosaic knockouts are achievable using MADM, this method relies on the inefficient inter-chromosomal recombination events that require cell division[19]. The MADM and Red2cDNA systems both have reproducible initial labeling ratios between the different colors; however, quantitative analysis is not possible with the ifgMosaics system, as it yields variable labeling ratios (Supplementary Table 1).

While mosaic overexpression systems are useful for studying specific decisions regarding cell fate[18] and/or oncogenic activations[22], a mosaic knockout approach is more suitable for studying general effects on the system in various biological contexts. For example, conditional KO systems are necessary to study the role of essential or developmentally regulated genes with precise spatio-temporal control, and a tool that allows simultaneous genetic recombination and cell labeling in a mosaic fashion gives the opportunity to compare the effects of gene loss of function versus wild-type within the same tissue of the same animal, thus in a uniform genetic background.

In this study, we report a tunable and efficient mosaic knockout system: Red2Flpe-SCON. This system is based on the Confetti reporter system and uses red clone-specific Flpe expression together with our previously described FRT-based Short Conditional IntrON (SCON)[23]. Our system offers a versatile, tunable, and precise method to achieve Flp-based mosaic knockout with Cre-mediated multicolor labeling in mice. Combined with our growing repertoire of FRT-SCON modified alleles[23], Red2Flpe-SCON can be used to address the role of specific genes while fate-mapping wild-type and mutant cells in different biological contexts, from developmental studies to disease models.

## Results

### Generating Red2cDNA: a mosaic system based on the Confetti reporter allele using CRISPR nickase-mediated targeting

We previously reported the *Red2Onco system* − a series of modified Rosa26-Confetti (Brainbow2.1) alleles, which enable the mosaic, ectopic expression of oncogenes in a red fluorescent protein (RFP)-labeled, clone-specific manner[22]. We adapted an efficient electroporation protocol to achieve the desired gene knock-in, adjacent to the RFP, using CRISPR-Cas9 nickases with Confetti embryonic stem cells (ESCs); a process that usually takes approximately two to three weeks (Supplementary Fig. 1). The targeting approach harnesses the homology-directed repair (HDR) pathway, with homology arms of 600–700 bp in length. The Red2 targeting vector contains specific restriction cloning sites for inserting the cDNA sequence − downstream and in frame with the RFP and 2A peptide − and a PGK-Blasticidin-pA cassette with an inverted orientation for efficient antibiotic selection (Supplementary Fig. 1a). The RFP protein, *tdimer2*, of the confetti allele consists of two segments with repeated sequences. The left homology arm contains the sequence of one dimeric unit and can therefore be inserted to replace either dimer (with about 650 bp difference in length), where the second dimer is the desirable target[24] (Supplementary Fig. 1b). For the targeting experiment, we utilized a pair of sgRNAs with a Cas9-D10A nickase (which only cleaves strands that are complementary to the sgRNA) to minimize any potential off-target effects[25] (Supplementary Fig. 1c). Upon electroporation, a small fraction of cells showed

GFP expression from the Cas9 nickase vectors (Supplementary Fig. 1d). After 48–72 h of recovery, cells were subjected to blasticidin treatment to select for targeted clones (Supplementary Fig. 1e), which were checked by long range PCR genotyping. Verified clones were then injected into developing blastocysts to generate chimeras (Supplementary Fig. 1f). This CRISPR-mediated method of targeting enabled mosaic genetic Red2cDNA mouse lines to be generated efficiently.

### Red2-Flpe: a versatile tool that enables precise mosaic knockout with multicolor labeling

Here, we present a mosaic knockout system that involves the insertion of the Flpe recombinase sequence[26] adjacent to an RFP (tdimer2) linked with a P2A peptide. This setup allows for the RFP-labeled (red), clone-specific, Flpe-mediated recombination of the target FRT allele, while ensuring that all the other fluorescent proteins remain genetically unaltered (Fig. 1a, b). We named this system, *Red2Flpe*. The mosaic knockout system, induced upon Cre-mediated recombination, results in the presence of both wild-type and mutant knockout cells in the same tissue. The YFP-labeled wild-type cells and RFP-labeled mutant cells undergo clonal competition that involves the secretion of paracrine factors (Fig. 1b). Depending on how the target gene knockout impacts the clonal fitness, the mutant cells could either outcompete or become eliminated, correspondingly (Fig. 1b, c). Cells carrying cancer driver mutations are known to secret paracrine factors to suppress the growth of neighboring cells[22,27,28], Red2Flpe is therefore superior in accurately tracking the evolution of mutant cells and the wild-type cells in the same tissue with spatiotemporal resolution (Fig. 1c).

After activating *Red2Flpe* with the ubiquitous *Rosa-CreER^T2* line, we harvested the colon, pancreas, seminal vesicles, spleen, stomach and tongue (Supplementary Fig. 2a–f), and confirmed that both YFP and RFP were expressed in all of the analyzed tissues. Quantification of YFP+ and RFP+ clones showed no recombination bias for one fluorescent marker versus the other, as similar numbers of both clones were scored (Supplementary Fig. 2g, h). This demonstrated the utility of *Red2Flpe* in multiple organs and tissues with recombination rates that were suitable for mosaic lineage tracing studies.

We next tested whether the expression of Flpe on the modified Confetti allele could recombine with the FRT sites on different alleles in the mouse genome, in a red clone-specific manner. In combination with the intestinal epithelium-specific CreER^T2 line, *Villin-CreER^T2*, we crossed *Red2Flpe* with two separate FRT lines. Firstly, *Vil-CreER^T2;Red2Flpe* was crossed with a newly generated Apc-FRT (exon 15 flanked by FRT sites) (Fig. 2a); knockout of Apc leads to stabilized β-catenin accumulation and subsequent Wnt signaling activation[29], similar to the inactivation of other Wnt negative regulators[30,31]. One week after the Cre-mediated labeling (Fig. 2b), the *Vil-CreER^T2;Red2Flp;Apc^FRT/FRT* mouse intestine showed that the RFP clones, specifically, displayed elevated cytoplasmic β-catenin staining (Fig. 2c, d). Secondly, the *Vil-CreER^T2;Red2Flpe* line was crossed with an FRT-based GFP reporter line called *RCE:FRT* (an FRT-stop-FRT GFP reporter on the Rosa26 locus)[32] (Fig. 2e). We expected overlapping GFP and RFP signals but no overlap with any other colors, if Flpe worked efficiently and specifically. After one week of Cre induction, we only observed GFP expression in the Flpe-expressing RFP+ cells (Fig. 2f–h). The rate of accumulation of RFP/GFP double positive clones was comparable to other established Confetti-based reporters[8], showing a minimal delay between RFP expression and Flpe-dependent GFP expression. To assess the efficiency and specificity of Flpe/FRT recombination of Red2Flpe, primary intestinal cells were harvested and analyzed by flow cytometry at five, seven and ten days after tamoxifen injection (Fig. 2i–k). A stringent gate was set for the RFP-positive cells, as the maturation time is longer for tdimer2 (-120 min)[24] (Fig. 2i). We observed an increase in the proportion of GFP/RFP double-positive cells from day five (66.3%) to day seven (82.1%) and day ten

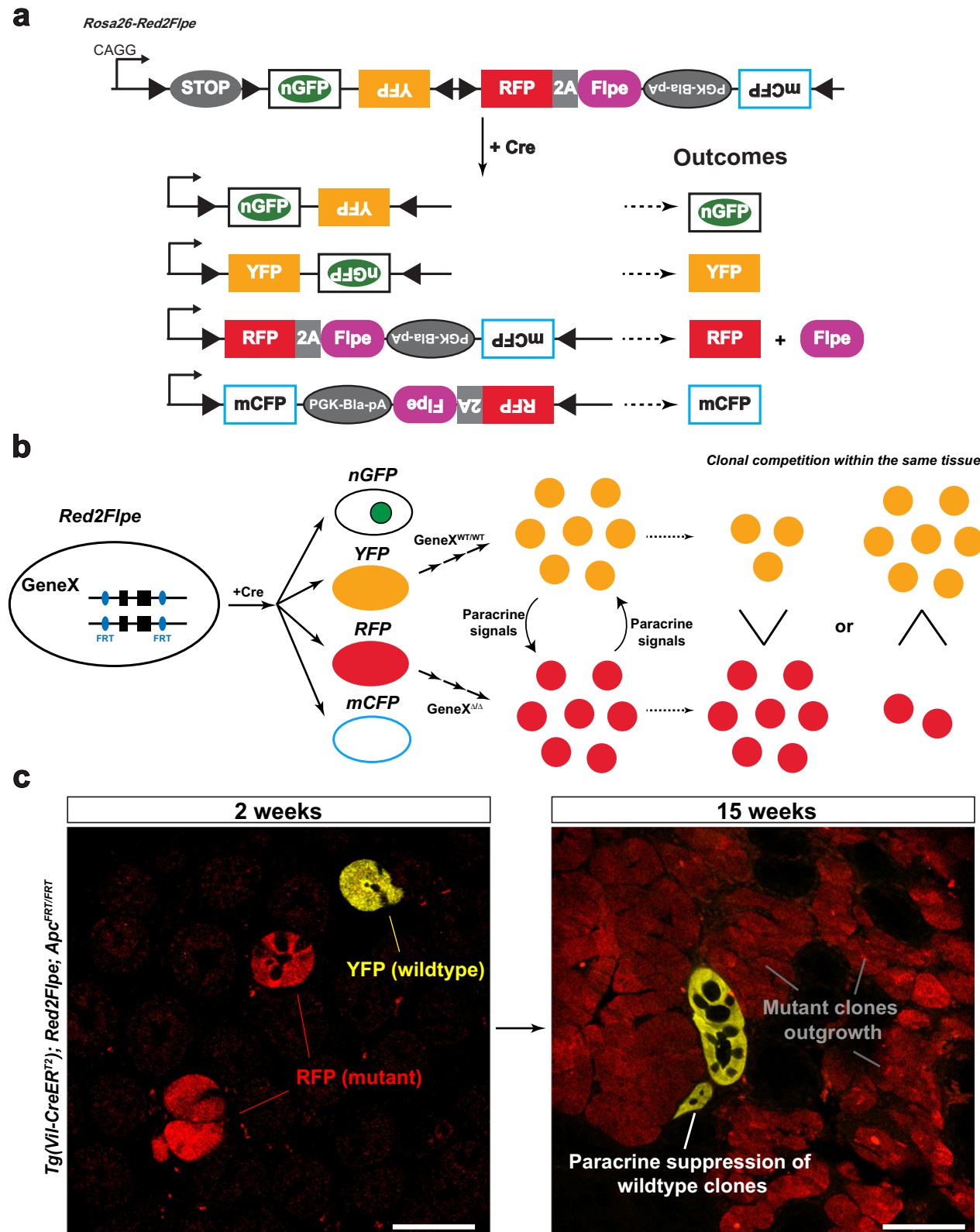

**Fig. 1 | Design of the Red2Flpe system: a mosaic knockout multicolor reporter allele. a** Upon Cre induction, each cell (unless it is polyploid) is adapted to express one fluorescent protein color. The cells labeled with red fluorescent protein (RFP) express the Flpe recombinase, while all the other fluorescent protein colors (green/yellow/cyan) correspond to wild-type cells. **b** Schematic showing the interactions between YFP+ wild-type cells (*GeneX^{WT/WT}*) and RFP+ mutant cells (*GeneX^{Δ/Δ}*) through paracrine signaling. As a result, the clonal competition between wild-type and mutant cells can be quantified. **c** Wholemount intestines of Tg(Vil-CreER^{T2}); Red2-Flpe; Apc^{FRT/FRT} mice from 2 and 15 weeks after tamoxifen administration. The mutant RFP cells, through the secretion of paracrine factors, outcompete the surrounding wild-type cells. The experiment in **c** was performed in 3 separate Tg(Vil-CreER^{T2}); Red2Flpe; Apc^{FRT/FRT} mice for each time point and the representative images were taken. Scale bar, 50 μm.

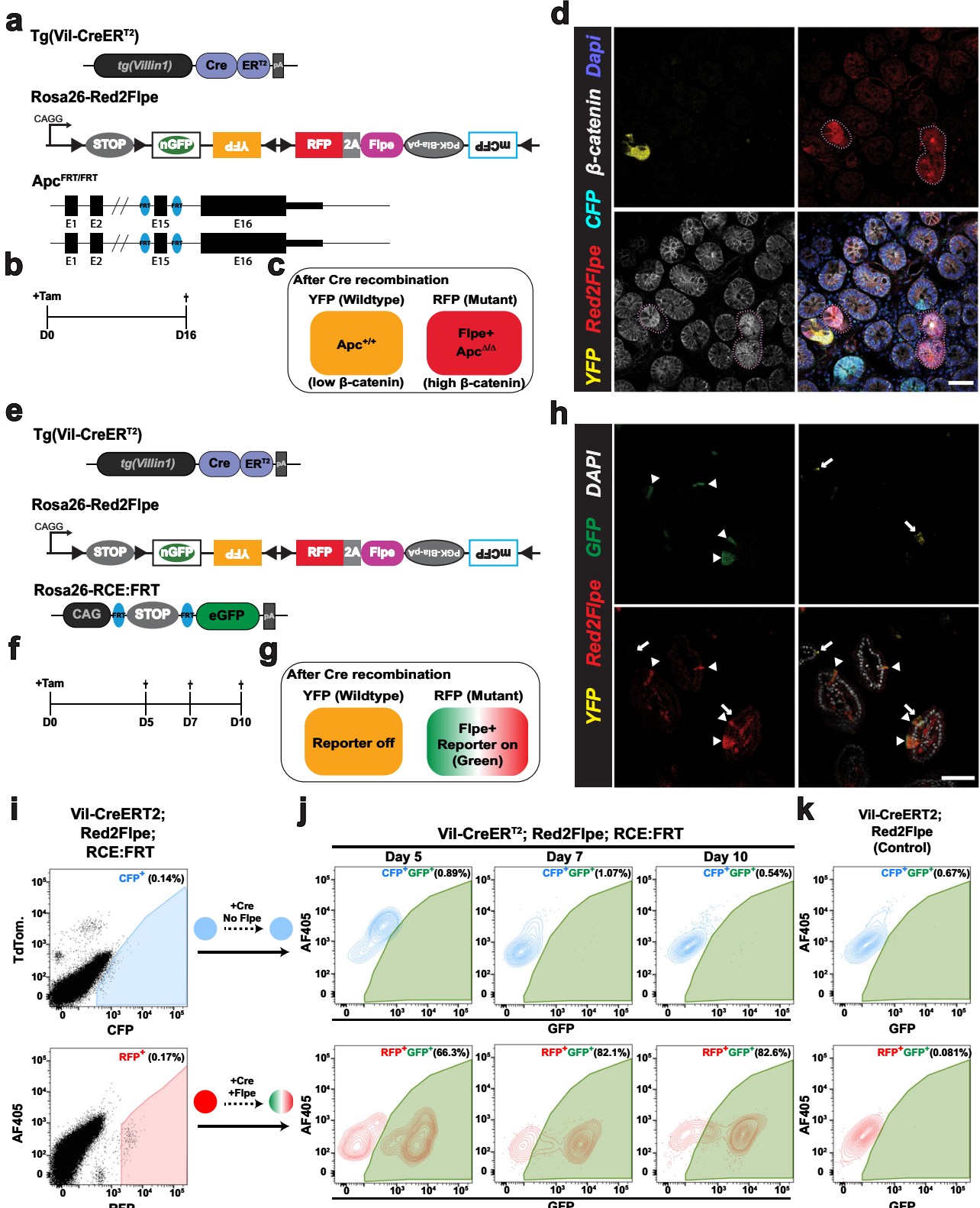

(82.6%) (Fig. 2j). There is a time lag between the point of tamoxifen injection, the observed Cre activity and the subsequent stable expression of the fluorescent reporters and Flpe recombinase; this data, therefore, suggests that Red2Flpe becomes active approximately five to seven days after the tamoxifen injection (Fig. 2j). Overall, these results indicate that *Red2Flpe* is a functional and efficient genetic tool that can be used to generate mosaic multicolor mouse models in vivo.

**Short Conditional intrON (SCON) facilitates the generation of FRT-based conditional knockout mouse lines with a one-step zygote injection**

As Cre-loxP was found to work more efficiently than the wild-type Flp, most of the murine conditional knockout (cKO) lines were made using the Cre-loxP system. Conditional mouse lines are often generated by in vitro ESC targeting, followed by a blastocyst injection to acquire

**Fig. 2 | Red2Flpe enables efficient red clone-specific recombination in vivo.**
**a**–**d** Small intestine samples harvested one week after treatment with tamoxifen (2 mg tamoxifen per 20 g body weight) from a Tg(Vil-CreER^{T2}); Red2Flpe; Apc^{FRT/FRT} mouse. The wild-type crypts labeled with either cyan (CFP), yellow (YFP), or no color show clear β-catenin staining in the membrane with less staining in the cytoplasm. The RFP-labeled (red) crypts display an increase in the cytoplasmic fraction of β-catenin staining − indicating successful knockout of *Apc* and malfunction of the β-catenin destruction complex. The RFP-labeled (red) crypts and the corresponding β-catenin staining are marked with dotted lines. The experiment in **a**–**d** was performed in 3 separate Tg(Vil-CreER^{T2}); Red2Flpe; Apc^{FRT/FRT} mice the representative images were taken after antibody staining. Scale bar, 50 μm. **e**–**h**, Small intestine samples harvested one week after treatment with tamoxifen (2 mg per 20 g body weight) from a Tg(Vil-CreER^{T2}); Red2Flpe; Gt(ROSA)26Sor^{tmL2(CAG-EGFP)Fsh} mouse. The expression of RFP (red) coincides with

GFP (green) but does not coincide with Confetti YFP (yellow). RFP-GFP double positive cells are marked by the triangles and the YFP cells are marked by the arrows. The experiment in **h** was performed in 2 separate Tg(Vil-CreER^{T2}); Red2-Flpe; Gt(ROSA)26Sor^{tmL2(CAG-EGFP)Fsh} mice the representative images were taken. **i** Gating strategy of YFP, RFP and CFP cells in tamoxifen-induced Tg(Vil-CreER^{T2}); Red2Flpe intestine. **j** GFP+ cells within the gated CFP+ population (top row) or the gated RFP+ population (bottom row) in an FRT-based reporter (Tg(Vil-CreER^{T2}); Red2Flpe; Gt(ROSA)26Sor^{tmL2(CAG-EGFP)Fsh}) mouse harvested 5, 7 or 10 days post tamoxifen administration. **k** GFP+ cells in the control mouse (Tg(Vil-CreER^{T2}); Red2Flpe) after tamoxifen administration. For each time point in **i** and **j**, the intestines from 2 mice of Tg(Vil-CreER^{T2}); Red2Flpe; Gt(ROSA)26Sor^{tmL2(CAG-EGFP)Fsh} and one control mouse of Tg(Vil-CreER^{T2}); Red2Flpe were profiled and showed similar results.

---

chimeric mice, and final germline transmission; this process takes at least six months. Despite efforts to improve the efficiency of the zygote injections, using CRISPR-based knock-in with a long single-stranded oligonucleotide (ssODN)[33], the process remains technically challenging.

To facilitate cKO mouse generation, we recently developed an artificial intron-based approach that uses a Short Conditional intrON (SCON), which is just 189 bp in length[23]. This method only requires a synthesized oligo template, a synthesized gRNA, and a commercially available Cas protein and/or mRNA. All these components are injected into zygotes to generate a cKO. SCON has a neutral effect following the initial insertion of the target gene, and induces the expected loss of function effect upon recombination in vivo. SCON, therefore, offers an alternative but efficient way to generate cKO alleles using a method that is as simple as CRISPR-based 'tagging'.

We reasoned that the loxP recombination sites used with SCON could be exchanged with FRTs, for compatibility with the Red2Flpe mosaic knockout system. The SCON-FRT system works in the same way as the SCON-loxP system − which consists of a splice donor, branch point, and splice acceptor − with SCON acting as a functional intron. With this system, two FRT recombination sites flank the branch point; the removal of the branch point upon Flp-mediated recombination then abrogates the SCON's intronic function, causing it to be retained in the mature transcript after splicing. The remaining 55 bp SCON intron sequence contains potential stop codons that can cause premature termination of translation and subsequent truncation of the target protein (Fig. 3a).

We next tested whether the SCON-FRT system had a similar neutral effect on gene expression, as with SCON-loxP. We transfected HEK293T cells with either intact eGFP; eG-SCON-FRT-FP (eGFP with a SCON-FRT insertion); or the recombined form, eG-SC-FRT-FP (Fig. 3b). We found that the intact eGFP and the two different eG-SCON-FRT-FPs, with either wild-type or F3 FRT sites, had comparable GFP levels; however, the wild-type FRT version slightly out-performed the F3 FRT version (Fig. 3c). With the recombined forms, the level of GFP fluorescence was not detectable, indicating loss of expression (Fig. 3b, c). We also tested whether SCON-FRT could be efficiently recombined in mammalian cells upon Flp expression. We generated mouse ESCs with constitutive expression of eG-SCON-FRT-FP, using the piggyBac transposon system. After transfecting an Flp-expressing plasmid, we found that the level of GFP diminished significantly, which confirmed the compatibility of the SCON-FRT for Flp/FRT-based recombination system in mammalian cells (Fig. 3d). Therefore, we concluded that the SCON-FRT system was also neutral, like the SCON-loxP system, and it was applicable in mammalian cells.

### Red2Flpe can be efficiently utilized in combination with the SCON-FRT system
Next, we assessed the compatibility of the SCON-FRT system with *Red2Flpe*. We made use of *Confetti* and *Red2Flpe* ESCs in combination with piggyBac-eG-SCON-FRT-FPs (Fig. 4a, b). Cells with an integrated

eG-SCON-FRT-FP were selected with puromycin with the expectation that both eGFP and puromycin-resistance expression would be compromised following Flp-mediated FRT recombination (Fig. 4a, b). After selecting GFP-expressing cells, we induced the recombination of the *Confetti* and *Red2Flpe* alleles, respectively, by transfecting a Cre-expressing plasmid. Using fluorescent activated cell sorting (FACS), RFP+ cells were sorted and cultured separately from the uninduced cells. All the RFP+ colonies of *Red2Flpe* ESCs showed no eGFP expression compared to the *Confetti* ESCs, which were used as controls (Fig. 4c). Flow cytometry revealed that most of the cells in the uninduced cultures retained high levels of GFP expression, despite some transgene silencing, while all of the recombined RFP+ cells lost GFP expression completely (Fig. 4d). These results indicated that Red2Flpe could be coupled with SCON-FRT to successfully achieve conditional mosaic gene knockouts.

### Sox2-SCON-FRT mouse generation via one-step zygote injection
With the knowledge that the use of SCONs would not affect basal gene expression, we generated thirteen cKO mouse lines[23]. We injected CRISPR-Cas9 ribonucleoprotein (RNP), Cas9 mRNA, and a 300 bp long ssODN of SCON (using either loxP sites or FRT sites, both of which were 189 bp long). We used left and right homology arms that were 55 and 56 bp long, respectively[23]. With the SCON approach, a complete experimental mouse line for mosaic knockout studies could be generated quickly and easily.

We propose a pipeline for generating zygotes using SCON targeting from the desired *CreER* and *Red2Flpe* lines, both in homozygosity (Supplementary Fig. 3a). From the offspring, pups that are heterozygous for the SCON-FRT knock-in can be used for mating to create further experimental lines (Supplementary Fig. 3b). The chance of acquiring the desired experimental cohort is 1/4 (Supplementary Fig. 3c) so, theoretically, it is possible to generate SCON-based *Red2Flpe* mosaic knockout mice by zygote injection with just two mouse generations.

One of the first SCON-FRT lines we generated was the *Sox2-SCON-FRT* (*Sox2^{scon}*) line (Supplementary Fig. 4a). From twenty founder pups, we obtained three heterozygous SCON-FRT knock-in mice with precise integration (15% efficiency). *Sox2^{+/scon}* mice can be bred to be homozygous without any noticeable developmental defects (refer to Fig. 4g, h in Wu et al.)[23], which validates the utility of SCON-loxP and FRT for in vivo experiments. The mouse ESCs derived from homozygous *Sox2^{scon/scon}* blastocysts expanded stably in culture. Then, following transient Flpe expression, the colonies that contained either mosaic or complete Sox2-KO cells exhibited distorted morphologies. These results were consistent with the importance of Sox2 in maintaining pluripotency (Supplementary Fig. 4b)[34].

### Mosaic knockout of Sox2 in adult tissues reveals its variable essentiality
*Sox2* is a transcription factor that plays crucial roles during embryonic development − from the blastocyst stages to the fate-specifying stages

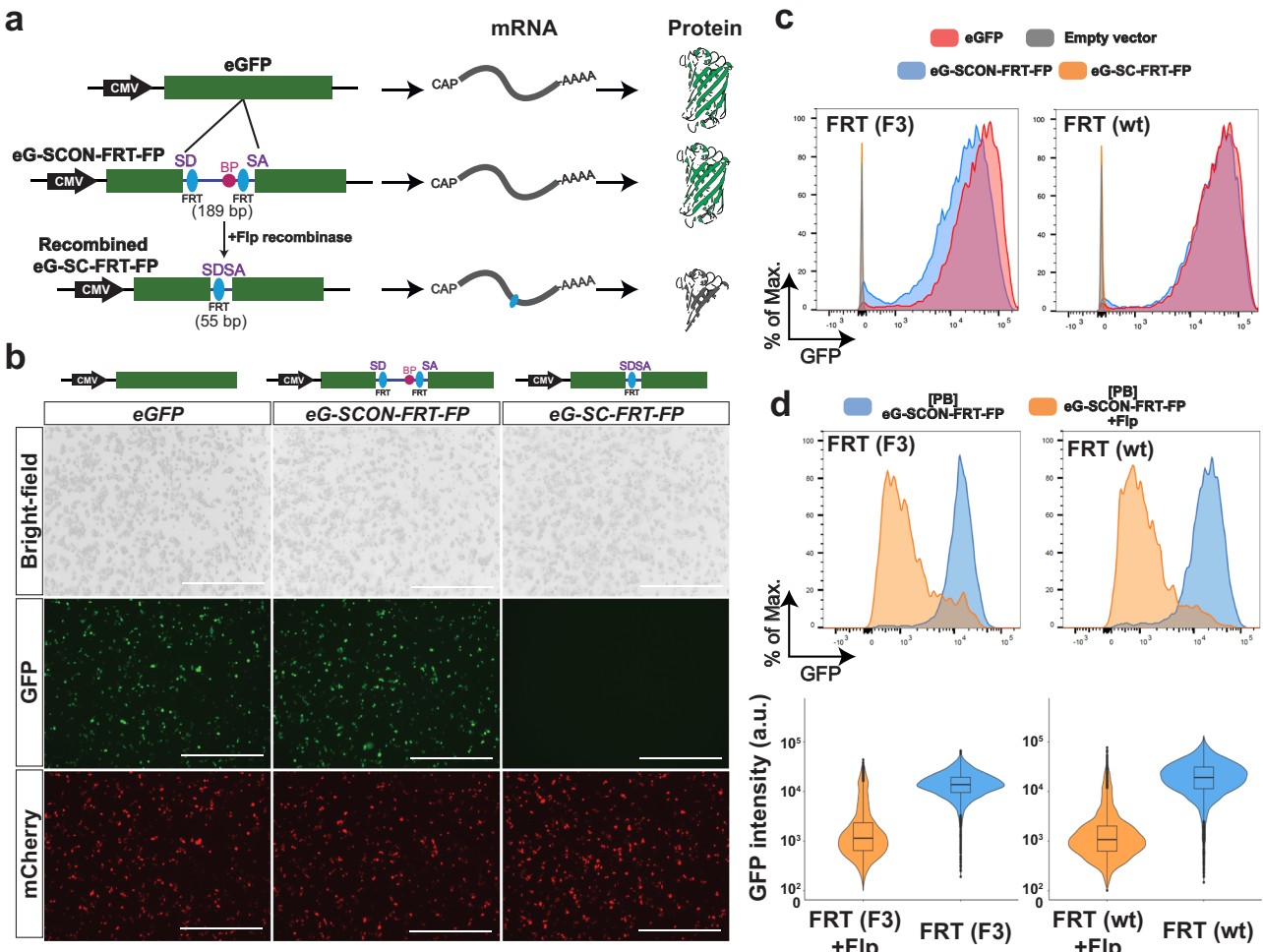

**Fig. 3 | SCON-FRT is a Flpe/FRT-based conditional knockout system.**
**a** Schematic diagram of SCON-FRT in an eGFP overexpression construct, including eGFP, eG-SCON-FRT-FP and recombined eG-SC-FRT-FP. SD (splice donor), BP (branch point), SA (splice acceptor). **b** Brightfield and fluorescent images of HEK293T cells 24 h after transfection. **c** GFP fluorescence level of HEK293T cells 48 h after transfection. Red: intact eGFP; Blue: eG-SCON-FRT-FP; Orange: recombined eG-SC-FRT-FP. **d** GFP fluorescence level of HEK293T cells with integrated eG-SCON-FRT-FP before (blue) and after (orange) transfection of a Flp-expressing plasmid. Experiments in **c** and **d** were performed twice and in two separate cell clones, which showed similar results. Violin and box plots indicate the distribution of data where minimum and maximum values are presented. The boxplot represents the central 50% of data points and the thickened line marks the median value. Source Data relevant to this Figure are provided with this paper in Source Data file.

− of many tissues[35]. Knockout studies revealed that *Sox2* is required for proper development of the esophagus[36,37]. In adults, *Sox2* is expressed in the esophagus and stomach, and is thought to be important for stem cell maintenance[38,39]. Consistently, tissue-wide knockout of Sox2 or depletion of Sox2-expressing cells leads to compromised tissue maintenance and physiology[38,39]. However, as widespread knockout of Sox2 in these tissues compromises their overall integrity, the exact function of Sox2 remains unclear. A mosaic analysis of Sox2 in the adult esophagus and other tissues is therefore necessary.

We used *Red2Flpe* and the *Sox2^{scon}* alleles with the ubiquitous *Rosa26-CreER^{T2}* inducer line, and generated *Rosa26-CreER^{T2}; Red2Flpe; Sox2^{scon/scon}* (*Red2Sox2KO*) mice to investigate the function of Sox2 in the adult mouse esophagus (Fig. 5a). By administering tamoxifen to the *Red2Sox2KO* mice, we activated fluorescent labeling and the red clone-specific knockout of Sox2 (Fig. 5b). We then lineage traced the wild-type (yellow) and knockout (red) cells and quantified the sizes of their clones over time. If Sox2 was essential for stem cell maintenance, we would have expected that the mutant clones would be rapidly lost from the basal layer within a short period of time. By contrast, if Sox2 was not essential for stem cell maintenance, we would have expected that the mutant clones would remain present in the basal layer − and might only be lost due to their relative clonal fitness in that tissue.

We collected the esophagus samples at different time points following the tamoxifen injection (Fig. 5c). We first confirmed that Sox2 expression was absent from the RFP+ clones in the esophagus at two weeks and four weeks, and checked that the YFP+ clones expressed Sox2, as expected (Fig. 5d). Interestingly, the Sox2 knockout mutant clones remained in the tissue, even after long-term tracing, which suggested that Sox2 might not be essential for stem cell maintenance in the esophagus (Fig. 5e, f). At 2.4 weeks after induction, the RFP+ and YFP+ clones were found at similar levels (Fig. 5f). However, at 4 and 9 weeks, the number of RFP+ clones was reduced compared with the wild-type YFP+ clones (Fig. 5f). We also quantified the clone sizes, and the location of the labeled cells within each clone − noting whether they were in the basal, parabasal (stratifying) or suprabasal layers. We found that the size of the RFP+ clones was consistently smaller than the size of the YFP+ clones (Fig. 5g). In addition, the RFP+ clones mainly consisted of cells in the basal and parabasal layers, whereas the cells in the YFP+ clones were more evenly distributed across all three layers (Fig. 5g, h). This suggests that, while the wild-type cells were able to efficiently self-renew and differentiate, the Sox2 mutant cells had reduced capacity for proliferation and differentiation. This eventually led to the loss of RFP+ clones from constant niche competition due to a reduction in their fitness compared to the wild-type cells.

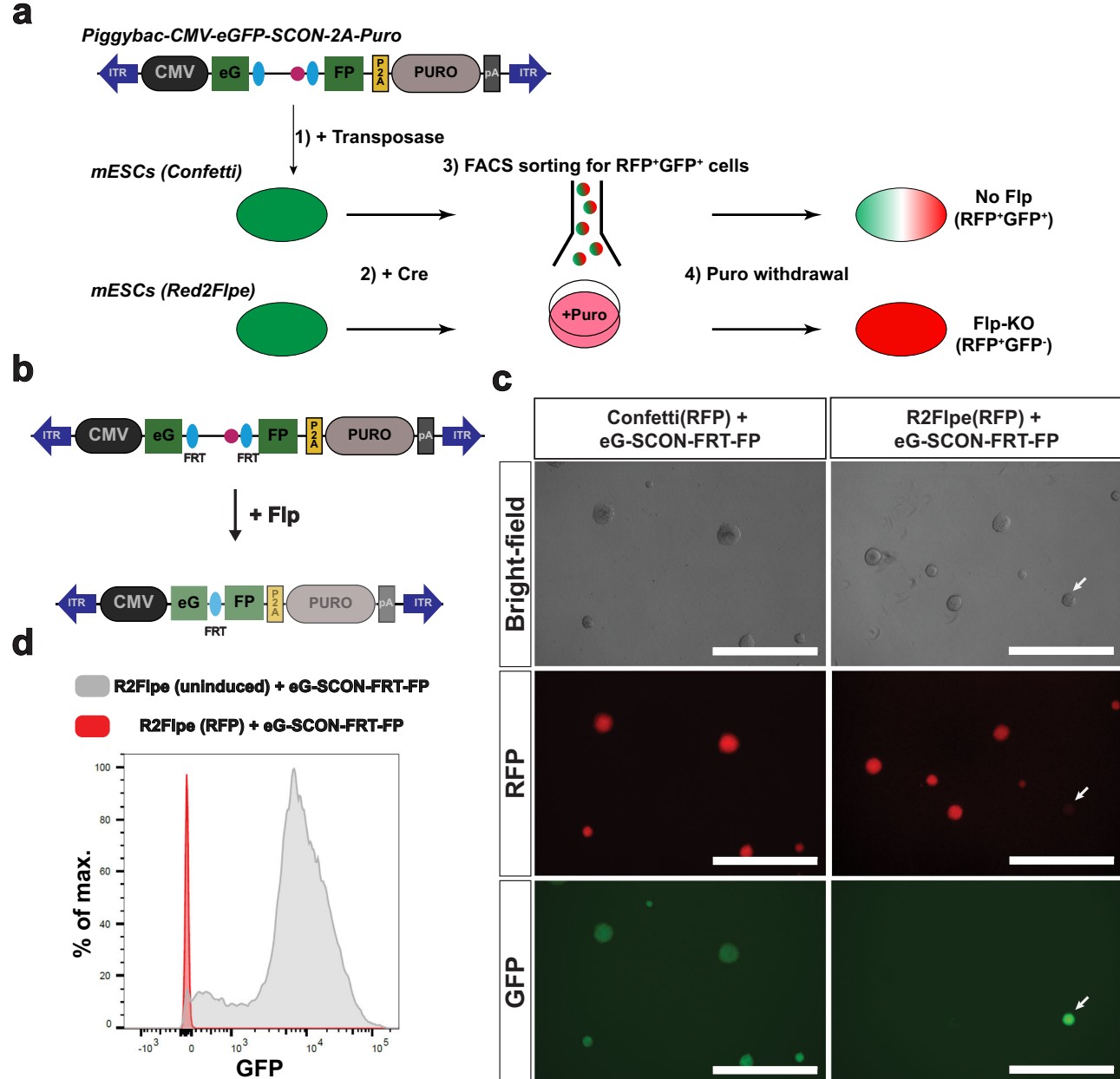

**Fig. 4 | Red2Flpe is compatible with the SCON-FRT system in vitro. a** Schematic of the experimental set up that involves in integration of a piggybac-eG-SCON-FP vector into the *Confetti* or *Red2Flpe* embryonic stem cells. Upon Cre recombination, the RFP + GFP+ cells are sorted and cultured in puromycin-containing media. To assess the recombination, and the loss of GFP signals, puromycin was omitted from the media. **b** Schematic of the piggybac vector carrying the overexpression cassette of eG-SCON-FRT-FP coupled with puromycin resistance. Both eGFP and puromycin

expression are reduced following Flp/FRT recombination. **c** Brightfield and fluorescent images of RFP+ Confetti and Red2Flpe ESCs with integrated eG-SCON-FRT-FP. The uninduced clone that retains eGFP expression is marked by an arrow. **d** GFP fluorescence level of Red2Flpe ESCs with integrated eG-SCON-FRT-FP after puromycin withdrawal. Gray: uninduced Red2Flpe; eG-SCON-FRT-FP cells; Red: RFP+ cells. Experiment shown in **c** and **d** was performed twice in two separate cell clones, which showed similar results.

We also utilized the esophageal organoid system to examine the behavior of Sox2 mutant clones over time. We found that the organoid-forming efficiency was slightly lower but maintained (especially in WENR+Nic condition) in Sox2 mutant cells compared to wild-type cells, with clearly smaller organoid sizes observed for all three culture medium conditions tested (WENR+Nic, ENR, or EN) (Supplementary Fig. 5). Nonetheless, Sox2 mutant organoids could still be passaged with both WENR+Nic and ENR media, suggesting persistent stem cell activity. Consistent with the in vivo lineage tracing results above, the esophageal cells that were depleted for Sox2 still retained their stem cell characteristics but exhibited reduced capacity for proliferation and differentiation. We also performed single cell RNA

sequencing (scRNA-seq) of wild-type (YFP+) and the Sox2-mutant (RFP+) cells (Supplementary Fig. 6). To overcome the low induction rate and unequal cell representation, we utilized a 384 plate-based sorting method followed by scRNA-seq called SORT-seq[40] to obtain comparable number of wild-type and mutant cells. We obtained 284 YFP+ cells and 303 RFP+ cells from 3 different mice 4 weeks after tamoxifen injection (Supplementary Fig. 6a). Cells were clustered based on expression of gene markers and the three major cell types[41] (proliferating basal, quiescent basal and suprabasal) could be identified (Supplementary Fig. 6b, c). Wild-type (YFP+) and Sox2-mutant (RFP+) cells could be found in all three cell types (Supplementary Fig. 6b), which is in-line with our imaging results (Fig. 5). Intriguingly,

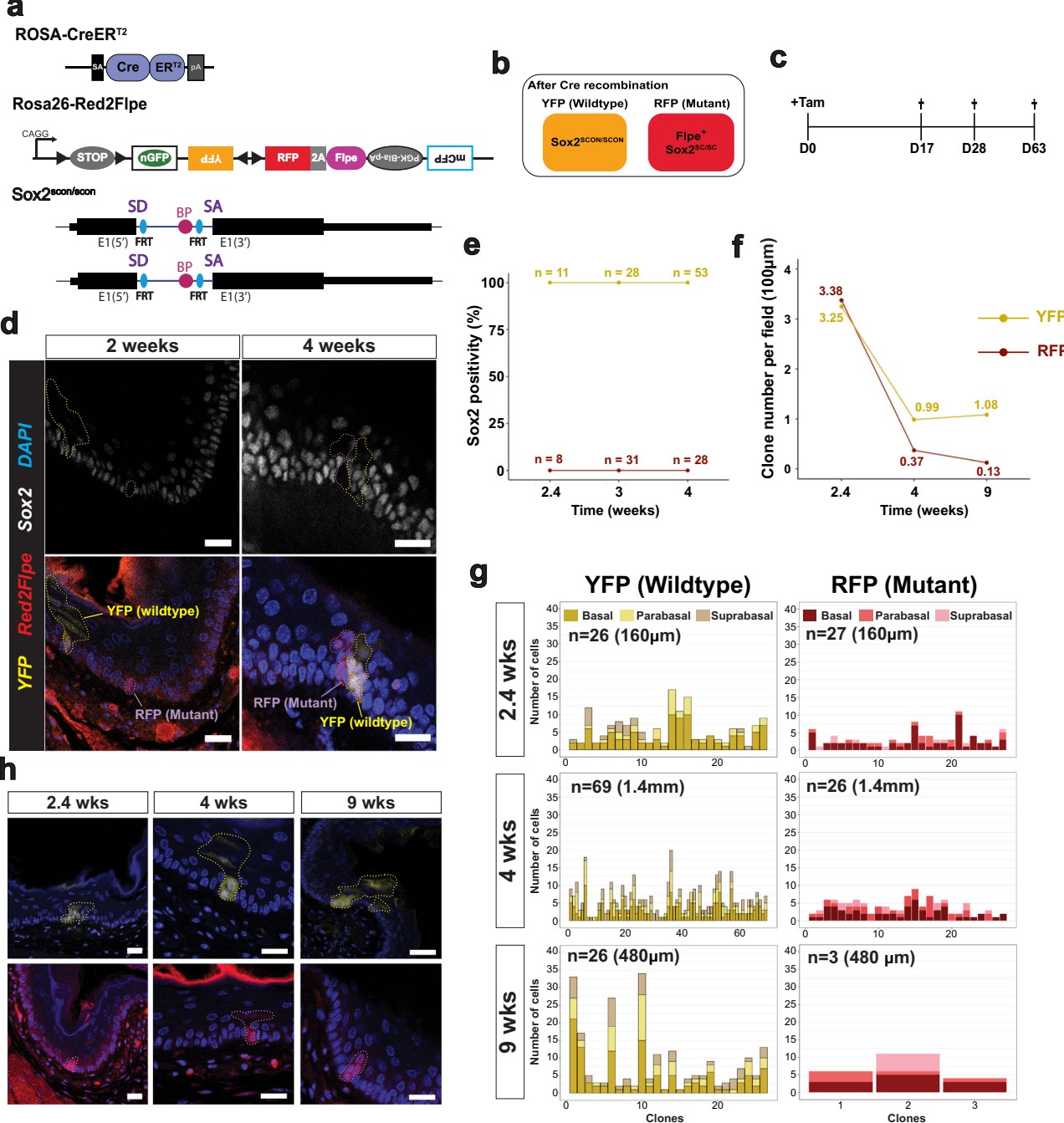

**Fig. 5 | Mosaic knockout of Sox2 reveals its role in the adult esophagus.** **a** Experimental set up for mosaic *Sox2* knockout in an adult mouse esophagus using the *Rosa-CreER[T2]; Red2Flpe; Sox2[scon/scon]* strain. **b** Schematic of the genotype of YFP and RFP cells after Cre administration. **c** A single dose of 3 mg tamoxifen was injected and the esophagus samples were then harvested on days 17, 28 and 63 after the injection. **d** Sox2 staining in esophageal sections at 2- and 4-weeks post-induction. The YFP and RFP clones are marked by dashed lines. Scale bar, 20 μm. **e**, **f** Quantification of Sox2+ cells and number of YFP+ and RFP+ cells per field of 100 μm at 2.4-, 4- and 9-weeks post-induction. **g** Quantification of spatial distribution of YFP+ and RFP+ clones at 2.4-, 4- and 9-weeks post-induction. **h** Representative images of YFP+ and RFP+ clones at 2.4-, 4- and 9-weeks post-induction. The YFP+ and RFP+ clones are marked by dashed lines. Experiments in **c**–**h** were performed in two mice for each time point, in which multiple fields of sectioned esophagus were imaged and quantified. Scale bar, 20 μm. Source Data relevant to this Figure are provided with this paper in Source Data file.

YFP+ and RFP+ cells did not show significant differences across all three cell types based on marker gene expression levels[41], with the exception of *Krt14* and *Krt5* in the quiescent basal and suprabasal layers, respectively (Supplementary Fig. 6d). Sox2 is expressed in the developing foregut and remains expressed in the epithelium of the stomach as well as the esophagus[36]. Therefore, we also checked whether knockout of Sox2 would have an impact on clonal fitness in

the stomach epithelium. We found that both the wild-type (YFP+) and the Sox2-mutant (RFP+) glands showed comparable labeling four weeks after being induced (Supplementary Fig. 7a). We also observed the comparable presence of both wild-type and Sox2-mutant clones in the base (Supplementary Fig. 7b, c) and isthmus (Supplementary Fig. 7d, e) parts of the glands[11]. Overall, this indicated that although Sox2 may affect stomach patterning and cellular differentiation, as

previously observed[36,42], it does not appear to significantly alter stem cell maintenance or clonal fitness in the stomach epithelium.

## Discussion

With the current work, we have presented a technology to produce mosaic KO in mice, and shown some examples of its application. Mosaic knockouts are preferred for investigating gene function, as they provide a physiologically normal background for mutant cells; this is especially useful to model certain diseases such as cancer, where only a subset of cells harbor specific mutations. Importantly, these mutant cells are within a background of wild-type cells, and the competition between cell populations shapes the course of disease progression[5,6]. On the other hand, ordinary conditional knockout approaches are associated with risks of strong secondary and tertiary effects, which can obscure the function of the target gene. Another example where a mosaic KO approach may be preferred is the study of cell competition: our Red2Flpe-SCON system allows the knockout of specific genes while visualizing wild-type and mutant cells at the same time, labeled with different fluorescent proteins, allowing the quantification of cell population dynamics, as we have shown here for Sox2.

Intriguingly, when we knocked out Sox2 in a mosaic manner in adult mice we found that Sox2 is not essential for long-term clonal maintenance in the epithelium of the esophagus and stomach. Sox2 is widely known as a stem cell transcription factor in ESCs; however, in the adult esophagus, Sox2 has a more specific role: regulating the proliferation and differentiation of basal cells. Based on the quantification of Sox2 mutant cell behaviors, we found that loss of Sox2 did not abrogate stemness completely, and clonal loss was mainly due to the reduced fitness during cell-cell competition. This observation was confirmed by in vitro organoid culture, where Sox2-mutant esophageal organoids could still be maintained, despite a reduced capacity for proliferation. In addition, the effects of Sox2 mosaic KO in the stomach epithelium were even less evident, as RFP+ mutant clones were almost indistinguishable from YFP+ wild-type clones. The reason for this difference in Sox2 KO susceptibility between stomach and esophagus may lie in the fact that high levels of Sox2 mark terminally differentiated enterochromaffin cells in the stomach[42]. Instead, isthmus-like stem cells express low levels of Sox2 and its absence may be compensated by other factors[42].

Despite significant technical improvements, the generation of cKO mouse lines by zygote injection remains challenging due to the requirement for long template insertion. With our SCON method, we generated several cKO lines using cloning-free reagents and a one-step zygote injection for both Cre-loxP and Flp-FRT recombination systems[23]. SCON alleles are functional in vivo and are compatible with the *Red2Flpe* line for mosaic knockout lineage tracing in the mouse. Considering the growing list of SCON alleles, as well as conventional FRT-flanked alleles already available, it will be possible to apply our technology to a wide number of genes.

In summary, we offer Red2Flpe-SCON as an efficient mosaic genetics tool for studying both wild-type and mutant cell behavior in the same tissue, while minimizing secondary effects. As this system is based on the widely used *Confetti* allele, versatile, tunable, precise, Flp-based mosaic knockouts with multicolor labeling can now be achieved in mice (Supplementary Table 1).

## Methods

### Ethical statement

All animal experiments were performed according to the guidelines of the Austrian Animal Experiments Act; with valid project licenses approved by the Austrian Federal Ministry of Education, Science and Research; and monitored by the institutional IMBA Ethics and Biosafety department. All animals were housed with 2–5 littermates based on litter size with food and water ad libitum. All animals were maintained at $22 \pm 1\,°C$, 50% relative humidity, and on a 12-h light/dark cycle).

### Mice

In this study, Red2Flpe and Apc-FRT mice were generated via embryonic stem cell (ESC) targeting and blastocyst injection. The Red2Flpe mouse was made by targeting in Confetti ESCs of C57BL6/129F1 background, and the Apc-FRT mouse with exon 15 flanked by FRT sites was generated by targeting in ESCs of B6129SF1/J background. Clonal-derived ESCs were injected into C57BL6 blastocysts. Chimeras were then backcrossed to C57BL6 to confirm germline transmission. See also Supplementary Table 2 for sequences of oligos and primers.

### Administering Tamoxifen and harvesting organs for tissue imaging

A dose of either Tamoxifen (Sigma, T5648) dissolved in corn oil (Sigma, C8267), or corn oil only, was injected intraperitoneally into 8–12-week-old mice, with a final concentration of 3 mg tamoxifen per 20 g of body weight. Esophagus, stomach, colon, pancreas, spleen, tongue and seminal vesicle samples were harvested, cleaned with cold PBS, and then fixed with 4% paraformaldehyde overnight (16 h) at 4 °C. The samples were then washed with cold PBS with 2–3 h intervals. Intestine samples were embedded in 4% low melting point agarose (Sigma, A9414) and then sectioned using a LAICA VT 1000S vibratome to produce 150-µm thick sections. Esophagus, stomach, spleen, tongue and seminal vesicle samples were incubated in 30% sucrose overnight at 4 °C, and then frozen on dry ice in O.C.T. solution (Scigen, 4586). The frozen section blocks were cryo-sectioned at 140 µm and then placed in PBS for further processing.

### Immunohistochemistry

Staining was carried out according to methods previously described by Yum et al.[22]. Briefly, samples were incubated in a blocking solution (5% DMSO, 0.5% Triton X-100, and 2% normal donkey serum (NDS) in PBS) for 4 h at 4 °C on a shaker. The solution was then exchanged with primary antibody diluted in a staining solution (1% DMSO, 0.5% Triton X-100, and 2% NDS in PBS): Sox2 (1:200; Invitrogen, 14-9811-82) or AF647-conjugated β-catenin (1:200; Cell Signaling Technology, 4627S). Sections were then incubated for 48 h at 4 °C on a shaker. After incubation, the samples were washed three times in cold PBS with 2–3 h intervals between each wash. The samples were then stained with secondary antibodies (1:500; donkey anti-rat AF647, Invitrogen) diluted in staining solution and incubated for 48 h at 4 °C on a shaker. The samples were then washed in PBS three times and stained with DAPI overnight at 4 °C. After washing, the samples were transferred to microscopic slides and mounted in RapiClear1.52 (Sunjin Lab). All confocal images were taken using a multiphoton SP8 confocal microscope (Leica).

### Preparation of primary intestinal cells for fluorescence-activated cell sorting (FACS)

The proximal halves of the freshly isolated intestines were gently cleaned by flushing them through cold PBS. The tissues were opened longitudinally and then shaken vigorously in cold PBS to clean the samples further. Once the solution became clear, the samples were incubated in Gentle Cell Dissociation Reagent (100-0485, STEMCELL technologies) for 30 min on ice. The samples were then shaken vigorously to release the cells from the tissue. The solution was then filtered through a 70 µm filter, and then centrifuged at $300 \times g$ for 5 min at 4 °C. Following this, the cell pellet was resuspended in TrypLE Express Enzyme (12604013, Gibco™) and incubated at 37 °C for 5 min to dissociate the cell clumps into single cells. Cold PBS was then added to the single cell suspension. The cell suspension was then centrifuged at $300 \times g$ for 5 min at 4 °C. Following this, the resulting cell pellet was resuspended in FACS buffer (2% FBS, 2 mM EDTA in PBS) and filtered using 40 µm strainers. The cells were analyzed using a BD-LSRFortessa flow cytometer (BD), and the flow cytometry data were analyzed using FlowJo software (BD).

### eG-SCON-FRT-FP constructs

The eG-SCON-FRT-FP cassette was ordered from Genscript, and subsequently cloned into pcDNA4/TO, and a piggybac overexpressing the plasmid. The pcDNA4/TO-eG-SCON-FRT-FP vectors were then transformed into Flp-expressing bacteria (A710, Gene bridges) to obtain the recombined forms. The correct clones were confirmed by restriction digest and subsequent Sanger sequencing.

### Cell cultures and transfections

**HEK293T cells.** Human embryonic kidney (HEK) 293T cells were cultured in high glucose DMEM containing 10% fetal bovine serum (FBS, Sigma), 1% penicillin-streptomycin (P/S; Sigma, P0781) and 1% L-glutamine (L-glut; Gibco, 25030024).

**Mouse ESCs.** The mouse ES cell line AN3-12 was cultured and transfected as previously described by Wu et al.[23], when they tested the recombination compatibility of SCON-FRT. In brief, the AN3-12 ES cells were cultured in high-glucose DMEM (Sigma, D1152) supplemented with 10% FBS (Sigma), 1% P/S, 1% L-glut, 1% NEAA (Sigma, M7145), 1% sodium pyruvate (Sigma, S8636), 0.1 mM 2-mercaptoethanol (Sigma, M7522), and $10^3$ units/ml of mouse LIF (stock concentration: $10^6$ units/ml, Merck Millipore ESG1107). Confetti and Red2Flpe ESCs were cultured in cell culture-grade dishes pre-coated with 10% gelatin (Sigma, G1890), and then supplied with DMEM/F12 (Sigma, D6421), Neurobasal (Gibco, 21103049), N2 (Gibco, 17502048), B27 (Gibco, 17504044), 1% P/S (Sigma, P0781), 0.1 mM 2-mercaptoethanol (Sigma, M7522), 1% L-glut (Gibco, 25030024), 10 mM PD0325901 (Axon, 1408), 10 mM CHIR99021 (Axon, 1386), and $10^3$ units/ml of mouse ESGRO LIF (Merck Millipore ESG1107).

**Plasmid transfection to Hek293T cells.** Transfection and subsequent analyses were performed as previously described by Wu et al.[23]. Briefly, 500,000-750,000 Hek293T cells were seeded in 6-well plates and left to attach and grow overnight. Following this, 2.5 μg of DNA (1 μg of mCherry-expressing plasmid (Addgene, 72264), and 1.5 μg of either pcDNA4/TO-eGFP, -eG-SCON-FRT-FP, or recombined forms of eG-SCON-FRT-FP was mixed with 8 μl of polyethyleneimine (1 mg/ml; Polysciences, 23966). These solutions were then incubated at room temperature for at least 15 min before being added dropwise to the cells. The culture medium was exchanged after 8–12 h or after an overnight transfection. The cells were examined 24–36 h after transfection, under an EVOS M7000 microscope (Thermo Scientific) using brightfield, GFP and TexasRed filters. Then, 36–48 h after transfection, cells were dissociated into single cells for flow cytometry analysis, with a BD-LSRFortessa flow cytometer (BD). Data from the flow cytometry experiments were analyzed using FlowJo software (BD).

### Organoid cultures

**Establishment and maintenance of esophageal organoids.** The esophagus harvested from an 8–12-week-old mouse was cut open longitudinally and incubated in 50 mM EDTA for 5 min at 37 °C. The epithelial layer was carefully peeled from the stroma, minced into small pieces and then incubated for 10 min at 37 °C in 0.5 mg/ml of Dispase dissolved in DMEM (Sigma; D4818). The solution containing the esophageal pieces was diluted in PBS and pipetted vigorously using a P1000 pipette before being filtered using a 30 μm cell strainer. The cells were spun down at $300 \times g$ for 5 min and then resuspended in cold PBS. This step was then repeated before embedding the cells in basement membrane extract (BME-R1) (R&D Systems, 3433010R1). Droplets measuring 15 μl were placed in a 48-well plate (Sigma, CLS3548-100EA) and transferred to a 37 °C incubator for 3–5 min until the BME had polymerized. Cells were supplemented with WENR+Nic medium, which consisted of advanced DMEM/F12 (Gibco, 12634028) supplemented with P/S (1%; Sigma, P0781), 10 mM HEPES (Gibco, 15630056), Glutamax (1%; Gibco, 35050061), B27 (2%; Life

Technologies, 17504044), Wnt3 conditioned medium (Wnt3a L-cells, 50% of final volume), 50 ng/ml recombinant mouse epidermal growth factor (EGF; Gibco, PMG8041), 100 ng/ml recombinant murine Noggin (PeproTech, 250-38), R-spondin-1 conditioned medium (HA-R-Spondin1-Fc 293T cells, 10% of final volume), and 10 mM nicotinamide (MilliporeSigma, N0636). This medium was used for the first two passages and was then changed to ENR media. Organoids were passaged weekly by collecting BME droplets in DMEM/F12 and pipetting several times to break up the droplets. They were then spun down at $900 \times g$ for 3 min. The supernatant was discarded carefully using a pipette, and the organoids were then resuspended in TrypLE Express (Gibco). They were then incubated at 37 °C for 10–15 min. Following this, the solution was pipetted using a p200 pipette to facilitate dissociation. The cells were then examined under the microscope. An aliquot of the cell solution was then transferred to a new tube to obtain a 1:10 split ratio. This was spun down at $900 \times g$ for 3 min, and then embedded in BME.

**Treatment with 4-Hydroxytamoxifen.** After two or more passages, esophageal organoids, cultured from single cells in the presence of EGF, Noggin, and R-Spondin 1 (collectively known as ENR), were grown for four days to allow organoids to expand in size. Medium containing vehicle (ethanol) or 500 nM 4-OH-tamoxifen (Sigma, H7904) was added after BME polymerized. After 8 h, the medium was changed back to ENR, and then replenished every two days. One week after the 4-OH Tam induction, the organoids were dissociated into single cells using TrypLE Express (Gibco). Cells were then sorted using a Sony SH800 cell sorter, equipped with 405 nm, 488 nm and 561 nm lasers (Sony).

**Imaging.** The cultured organoids were examined under an EVOS M7000 microscope (Thermo Scientific) using brightfield, GFP and RFP filters.

**Esophageal cell harvest for single cell 384-well plate sorting and scRNA sequencing.** Mice carrying *Rosa-CreER^{T2}; Red2Flpe; Sox2^{scon/scon}* induced with vehicle or 3 mg tamoxifen per 20 g of body weight for 4 weeks were sacrificed by cervical dislocation and the esophagi were harvested. The esophagi were cut open longitudinally and incubated in 50 mM EDTA for 5 min at 37 °C. The epithelial layers were carefully peeled from the stroma using forceps and washed in cold PBS. The epithelia were then minced into small pieces using surgical blades. The pieces were collected into tubes containing 0.5 mg/ml of Dispase dissolved in DMEM and incubated for 10 min at 37 °C. To facilitate tissue digestion, the mixtures were pipetted up and down after 5 min of incubation using a P1000. The mixtures were diluted in PBS and pipetted vigorously using a P1000 pipette before being filtered using a 30 μm cell strainer. The cells were spun down at 300 g for 5 min at 4 °C and resuspended in cold PBS. This step was repeated for a total of 3 times before the cells were resuspended in FACS buffer containing 2% fetal bovine serum, 2 mM EDTA in PBS. Cells were sorted with a FACS Aria III machine (BD) at high purity mode into pre-chilled 384-well plates obtained from Single Cell Discoveries (SCD) containing primers and mineral oil (Sigma)[40].

After sorting, plates were snap-frozen on dry ice and stored at −80 °C. For amplification cells were heat-lysed at 65 °C followed by cDNA synthesis using the CEL-Seq2 protocol[43] using robotic liquid handling platforms Nanodrop II (GC Biotech) and Mosquito (TTP Labtech). After second strand cDNA synthesis, the barcoded material was pooled into libraries of 384 cells and amplified using IVT. Following amplification, the rest of the CEL-seq2 protocol was followed for preparation of the amplified cDNA library, using TruSeq small RNA primers (Illumina). The DNA library was paired-end sequenced on an Illumina Nextseq™ 500, high output, with a $1 \times 75$ bp Illumina kit.

**Single-cell RNA sequencing data analysis.** After Illumina sequencing, Read 1 was assigned 26 base pairs and was used for identification of illumina library barcode, cell barcode and UMI. R2 was assigned 60 base pairs and used to map to the reference transcriptome of mm10. Raw FASTQ files for scRNA-seq were processed by using the zUMIs (v2.9.7)[44] pipeline and reads were aligned to the mouse reference genome (mm10) using the STAR aligner (v2.7.3)[45] with the whitelist of cell barcodes and the fasta file of ERCC spike-in sequences were used. Based on the whitelist of cell barcodes and quality control metrics, cells containing valid barcodes with >2.5 log10-scaled counts of unique molecular identifiers (UMIs) and <50% of UMIs assigned to ERCC spike-in sequences were used for analysis by using the scater (v1.22.0)[46] R package. The raw UMI counts without ERCC spike-in counts were normalized with cell-specific size factors by using the scran (v1.22.1)[47] R package. Top 1000 highly variable genes (HVGs) were selected by using the scran R package. The principal components (PCs) were calculated with the scaled normalized counts with HVGs by using the RunPCA function of the Seurat (v4.0.5)[48] R package. Top 50 PCs were used for clustering and dimension reduction. Cell clusters were defined by using the FindNeighbors and the FindClusters function of the Seurat R package. The two-dimensional coordinates were calculated by using the RunUMAP function of the Seurat R package. Non-epithelial cells expressing Ptprc were excluded and data processing with remaining cells was repeated using the same methods described above. Cell-types were assigned by clusters with canonical marker genes for proliferating basal cells, quiescent basal cells and suprabasal cells. Label information for wild-type (YFP+) and the Sox2-mutant (RFP +) cells from different mice is provided as Supplementary Data 1 and 2.

**Statistics and reproducibility.** All experiments were repeated at least 2 times to confirm reproducibility, from which representative images were selected. FACS analysis and quantification of wild-type and mutant clones from mice were performed from 2 mice for each time point. No statistical method was used to predetermine sample size. No data were excluded from the analyses. The experiments were not randomized, and the investigators were not blinded to allocation during experiments and outcome assessment. The selected time points for lineage tracing and sample size of 2 mice per time point were chosen based on previous related lineage tracing studies in the GI tract[8,10,12,14,22].

### Reporting summary
Further information on research design is available in the Nature Portfolio Reporting Summary linked to this article.

## Data availability
The raw and processed scRNA-seq data are available in the gene expression omnibus (GEO) database under accession code GSE266542. All data needed to evaluate the conclusions in the paper are present in the paper and/or the Supplementary Materials. Source data underlying Figs. 3, 5 and Supplementary Figs. 2 and 6 are provided. Source data are provided with this paper as a Source Data file. Source data are provided with this paper.

## Code availability
Codes used for scRNA-seq data analysis are available at https://github.com/CB-postech/NATURE-COMMUNICATIONS-mEsophagus-SOX2.

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

## Acknowledgements

We thank present and past members of the Koo, Elling and Urban labs at IMBA for valuable discussions and critical comments, Dr. Rike Zietlow and the Life Science Editors for reading and correcting the manuscript, VBC core facilities (especially the Histopathology facility, BioOptics and the animal caretakers). We thank Single Cell Discoveries for helping us with single cell RNA sequencing and offering technical advice on sample handling. This work was supported by core funding from the Institute of Molecular Biotechnology (IMBA) of the Austrian Academy of Sciences; ERC starting grant, Troy Stem cells, 639050; Interpark Bio-Convergence Center Grant Program; and fellowship to S.W. (DOC Fellowship of the Austrian Academy of Sciences). G.C. was supported by a Lise Meitner Postdoctoral fellowship M2976, FWF, and by the FWF Standalone (P35694) and ERA PerMed (I 5900) grants. B.-K.K. and his team are supported by the Institute for Basic Science.

## Author contributions

B.-K.K., Joo-Hyeon L. and B.D.S. formulated the design of Red2cDNA. S.W. and B.-K.K. planned and designed the experiments. R.B. and S.W. performed SCON-FRT transfection experiments and the subsequent FACS analysis. Ji-Hyun L. and S.W. analyzed the stomach phenotype of mosaic Sox2 knockout. S.W. performed all experiments, with help from G.C., S.Y.P., I.T., J.K., N.H. and S.P.C., and the core facilities at IMBA/IMP. C.T. performed blastocyst injection of targeted mouse ESCs to generate Red2Flpe and ApcFRT mice. S.K., H.L., and J.K.K. performed all computational analysis. S.W., G.C., B.-K.K. wrote the manuscript. S.W., G.C. and B.-K.K. completed the revision of the manuscript.

## Competing interests

S.W. and B.K.K. are inventors of a patent application on the SCON technology used in this study which was submitted by the Institute of Molecular Biotechnology to the European patent office (EP21172761) followed by a PCT application (WO2022/234086A1), entitled "Controlled gene expression methods and means". The remaining authors declare no competing interests.
