## [Peer Review File · Nature Communications]

REVIEWER COMMENTS

Reviewer #1 (Remarks to the Author):

Lineage tracing is an essential technique in studying cell behavior in vivo, but to date many experiments have suffered from the limitation of having to use different animals to track wild type and mutant cells. This is a significant draw back, as even within the same inbred strain, variation in genetic background can alter tissue dynamics and phenotypes. The ability to track wild type and mutant cells within the same animal overcomes such potential artefacts, provides intra-organ comparisons of mutant effects. This paper reports a versatile, inducible system to achieve recombination and labelling allowing wild type and mutant cells to be followed with lineage tracing.

The introduction might perhaps be better focussed on the biological problems that this system will address and the advantages that it will bring, rather than just dealing with older, inefficient systems like MADM.

The first part of the results section is a very technical description of CRISPR-nickase knock-in technology. Whilst of interest, this could be shortened, and the extensive details moved to the methods section. The main interest in the paper starts at line 105, with the description of the dual recombinase system that forms the core innovation. In focussing solely on their own interest in paracrine oncogene driven signalling, line 112-118, the authors surely miss the much wider applications of this technology, which applies to any wild type/mutant combination. These comments would be better made in the discussion.

In Figs 1 and 2 the system is shown to work, with the kinetics of induction being characterised. Labelling accumulates over 5-7 days, which is quite slow, but there is very little 'leak' in the absence of Tamoxifen, so the system meets the essential requirements for inducible lineage tracing.

A major potential issue with a Flpe based system is the lack of available lines carrying Frt conditional alleles. This is elegantly dealt with by the application of the recently developed SCON approach. One line is demonstrated, a conditional allele of Sox2. Deletion of Sox2 leads to reduced fitness in esophageal progenitor cells as assessed by a clonal lineage tracing study (Fig 5), but has little effect in gastric epithelium.

Overall this paper presents important advances in transgenic technology that will extend the application of lineage tracing. The paper could be improved by rewriting, with emphasis on making the applications of the technology more accessible. It's essential that the mouse strains are made available by depositing them in a facility such as JAX or EMMA at the time of publication with a manuscript of this type.

Reviewer #2 (Remarks to the Author):

In this paper, researchers firstly built a Red2Flp2 system that is similar to the Red2Onco system which they published in 2021. Then, based on the SCON technology published in 2022, they constructed Sox2scon-FRT/ scon-FRT mice line, by which this study revealed that Sox2 is not essential for adult esophagus stem cell maintenance itself, but rather for stem cell proliferation and differentiation.

This study follows the previously published Red2Onco system in 2021 and scon-loxp technology in 2022. The scon-FRT/scon-FRT mice lines and the Sox2scon-FRT/scon-FRT mice line had been mentioned in their 2022 article. This research team took full advantage of the recombinant randomness of the Confetti mice line, combined with Flp-frt for gene expression blocking, and finally performed mosaic analysis of mutant and wt cells in vivo by two recombinant enzymes combinations.

Here are some questions or concerns on the manuscript:

(1) Compared with the original Confetti mice line, does the Red2Flp show any influence on the expression of fluorescent proteins or lead to the preference of fluorescent proteins' expression? (as the RFP's signal was not high-quality in the whole paper, and the YFP signal in 2h was also not good); Is the ratio of RFP+ cells to YFP+ cells stable in Rosa-CreERT2; Red2Flp mice line under homeostatic condition?

(2) Line122 mentions "the utility of Red2Flpe in multiple organs and tissues with recombination rates that were suitable for mosaic lineage tracing studies". It is necessary to match their representation, multiple organs should be listed including (the lungs, heart, liver, stomach, thymus, large intestine, pancreas, kidney, etc.). Besides, since the recombination rate is mentioned, should the relative statistical results be shown?

(3) The efficiency of gene knock-out needs to be shown by statistical results (High beta-catenin RFP+/RFP+); Recombination efficiency in Figure 2h also needs to be shown more directly (GFP+RFP+/RFP+ and GFP+YFP+/ YFP+).

if we want to use the system reported in this paper, we need to use A-CreER+/Ki; Red2Flpe+/Ki; X scon/scon mice line. The efficiency of experiments mostly depends on two key points, one is the correspondence of RFP+ expression in Red2Flpe mice line with Flp actual working efficiency; the other

one is whether Xscon/ scon can fully stop gene expression in vivo. None of these is shown by data in the article.

(4) Why not show the staining and statistical results (GFP+Mcherry+/ Mcherry+) of the Figure 3d results? it can be more intuitive to understand the recombination efficiency of Flp to eG-SCON-FRT-FP.

(5) Red2Flpe has a very high recombination efficiency for SCON-Frt in vitro (as shown in Figure 4d, 4c). But how about the efficiency of ROSA-CreERT2; Red2Flpe; eG-SCON-Frt-FP in vivo? (GFP+RFP+/RFP+)

(6) It requires the zygote injection technology to build CreER+/Ki; Xscon/+ or Red2Flpe+/Ki; Xscon/+ mice line. It represents a significant limitation to promoting such techniques, as similar mice lines need to be built each time.

(7) In the last part, the single-cell sequencing or bulk-seq of mut RFP+ cells and wt YFP+ cells can figure out the genetic differences and support "the esophageal cells that were depleted for Sox2 still retained their stem cell characteristics but exhibited reduced capacity for proliferation."

(8) Why Sox2 exhibited a reduced capacity for esophagus stem cell proliferation but did not appear to significantly alter stem cell maintenance or clonal fitness in the stomach epithelium?

(9) Why are the n values of YFP and RFP so different in Figure 5e?

(10) The system built up in this paper is based on Rosa26-Confetti (Brainbow2.1) mice line. However, they just compare mut RFP and wt YFP. What is the advantage of this system with some mosaic systems using two colors? More random?

Reviewer #3 (Remarks to the Author):

In their study, Wu and colleagues have developed a versatile set of tools to efficiently generate mouse models for mosaic knockouts within a short period of time. The purpose of these models is to simultaneously trace the lineage and study the fate of mutant and wild-type clones. Previously, they

reported the development of the Red2Onco mouse model (Yum et al., Nature 2021), which is a confetti/brainbow-like model. In this model, cre activity stochastically induces the expression of CFP, GFP, YFP, or RFP, and the latter one along with the co-expression of an oncogene. This model enables the mosaic induction of RFP mutant clones and wild-type CFP, GFP, and YFP clones. In their current study, the authors introduce a new model called Red2Flpe, where they have replaced the oncogene with Flpe. This novel model allows for the Cre-mediated induction of Flpe activity in the RFP clones. Importantly, by combining this model with their recently developed SCON models (models in which Flpe activity induces the conditional loss of a gene; Wu et al., Exp Mol Med 2022), the Red2Flpe model enables the induction of RFP expression and concurrent loss of a (tumor suppressor) gene. This represents a significant advancement for the field as it allows for the investigation of the fate and behavior of mutant cells lacking (tumor suppressor) genes, enabling comparisons with wild-type cells.

To illustrate the great potential of their new model, the authors successfully labeled Apc mutant cells in the intestine and correlated them with β -catenin accumulation. Additionally, they demonstrated that Sox2 is not an essential factor for stem cell maintenance, but it plays a crucial role in the proliferation and differentiation capacity of the gastric epithelium.

This paper describes an impressive improvement in our current toolkit, with technically well-executed experiments and a remarkable tour-de-force. Moreover, the authors reveal interesting new insights into the role of Sox2 in stem cell potential in the esophagus and stomach. Overall, I highly recommend publishing this manuscript in Nature Communications, with the condition that the authors address the following minor points:

Minor Comments:

Extended Data Fig. 2: The authors demonstrate the utility of Red2Flpe in seminal vesicles, spleen, and tongue. However, it would be valuable to include other more commonly studied organs such as the skin and intestine.

In Figures 2b, 2c, and 2d: The authors convincingly demonstrate β -catenin stabilization in RFP+ cells after a week of Tamoxifen-mediated Red2Flpe induction. It would be beneficial for the field to explore the timing of β -catenin accumulation. Is there a correlation between the number of mutant cells and β -catenin upregulation? How does β -catenin stabilization evolve over time in this mouse model? In other words, could the authors provide further insights into the heterogeneity of β -catenin accumulation in RFP+ cells from day 5 to day 10?

Fig. 3D: Does the SCON-FRT system lead to complete loss of GFP or a significant reduction? If it results in complete loss, could the authors include a plot showing an experiment where cells did not express any

construct (negative control)? If there is partial loss of GFP, it would be helpful to report on this. In many experiments, partial loss may even be favorable, and it is important to know whether the SCON-FRT system leads to knockdown or knockout.

In Figures 2i, 2j, and 2k: The authors present a time series of flow cytometry to isolate GFP/RFP cells and demonstrate the long maturation time of the Tdimer2 fluorophore. However, in Figure 2h, not every RFP cell is GFP positive. Could it be possible that the gating strategy of their flow cytometry setup is too stringent? It would be helpful to include images at various time points (e.g., day 5, 7, and 10) and analyze the GFP/RFP ratio in each image.

Figure 5: In this section, the authors investigate the role of Sox2 in stem cell maintenance. Interestingly, they demonstrate that Sox2 is not crucial for stem cell maintenance, as RFP+ clones remained in the tissue even after long-term tracing. However, is the loss of RFP+ clones not dependent on the turnover rate of cells in the esophagus? Slow turnover may potentially result in a gradual loss of clones. Furthermore, the authors find that Sox2 mutant cells have a lower organoid-forming efficiency compared to wild-type cells. Organoid-forming capacity is often considered an indicator of stem cell potential in the field. Therefore, it might be advisable to temper the conclusion slightly regarding the lack of effect of Sox2 knockout on stem cell maintenance.

REVIEWER COMMENTS

Reviewer #1 (Remarks to the Author):

Lineage tracing is an essential technique in studying cell behavior in vivo, but to date many experiments have suffered from the limitation of having to use different animals to track wild type and mutant cells. This is a significant draw back, as even within the same inbred strain, variation in genetic background can alter tissue dynamics and phenotypes. The ability to track wild type and mutant cells within the same animal overcomes such potential artefacts, provides intra-organ comparisons of mutant effects. This paper reports a versatile, inducible system to achieve recombination and labelling allowing wild type and mutant cells to be followed with lineage tracing.

The introduction might perhaps be better focussed on the biological problems that this system will address and the advantages that it will bring, rather than just dealing with older, inefficient systems like MADM.

R: the reviewer raises a good point. We have added more about the use and benefit that our Red2Flpe system can bring, at the end of Introduction, lines 75-87.

The first part of the results section is a very technical description of CRISPR-nickase knock-in technology. Whilst of interest, this could be shortened, and the extensive details moved to the methods section.

R: While it is true that this is quite a technical part, we believe it is more appropriate to keep it in the Result section, as it contains a step-by-step description of our system and the whole Extended Data Fig 1 is mentioned thoroughly in the text. We are afraid that, if we moved it to the Methods section, this part would not receive the attention it deserves.

The main interest in the paper starts at line 105, with the description of the dual recombinase system that forms the core innovation. In focussing solely on their own interest in paracrine oncogene driven signalling, line 112-118, the authors surely miss the much wider applications of this technology, which applies to any wild type/mutant combination. These comments would be better made in the discussion.

R: We have now modified the discussion to include a wider envision of potential applications of our technology. We thank the reviewer for this valuable point.

In Figs 1 and 2 the system is shown to work, with the kinetics of induction being characterised. Labelling accumulates over 5-7 days, which is quite slow, but there is very little 'leak' in the absence of Tamoxifen, so the system meets the essential requirements for inducible lineage tracing.

R: the reviewer is right that cell labeling should be achieved in a relatively short time to properly study the clonal dynamics of stem cells. Previous work, for example, has shown that expansion of labelled clones (id est, the accumulation of monochromatic cell clones) occurs within 7-14 day time-frame (see Snippert et al., Cell 2010, pp 137-138 – PMID 34079126). In this study, Cre-mediated reporter gene expression was completed in 3-4 days, which includes consecutive activation of Cre-mediated recombination and recombination-mediated reporter gene activation. Technically, our system includes one more step, the activation of Flp-mediated recombination, between Cre activation and reporter gene expression. Thus, theoretically, our system takes more time to complete the intended fluorescent labelling and gene knockout. Based on what we see from the GFP reporter, it seems like we need to wait for at least 5 days and we believe that it is still useful for long-term lineage tracing in most studies. We have now stressed this point in the Result section, lines 146-147.

A major potential issue with a Flpe based system is the lack of available lines carrying Frt conditional alleles. This is elegantly dealt with by the application of the recently developed SCON approach. One line is demonstrated, a conditional allele of Sox2. Deletion of Sox2 leads to reduced fitness in esophageal progenitor cells as assessed by a clonal lineage tracing study (Fig 5), but has little effect in gastric epithelium.

R: We are happy to see that the reviewer has noted the utility of our SCON system. We are expanding the list of available "SCONed" mouse strains, in which many lines are made for Flp

recombination (PMID: 36494589), and we think that this, together with a few standard FRT-flanked alleles already available, will contribute to making Red2Flpe technology a useful tool to many labs. We have emphasized this point in lines 320-322 of the Discussion.

Overall this paper presents important advances in transgenic technology that will extend the application of lineage tracing. The paper could be improved by rewriting, with emphasis on making the applications of the technology more accessible. It's essential that the mouse strains are made available by depositing them in a facility such as JAX or EMMA at the time of publication with a manuscript of this type.

R: We appreciate the enthusiasm of this Reviewer for our new system. We also appreciate the advice to improve our manuscript, indeed we have modified the introduction and discussion sections, putting more emphasis on how this new technology can be beneficial to the scientific community. The reviewer is right that our mouse lines should be made accessible. We will be contacting JAX for submitting our strains once the manuscript is published, as recommended by JAX policies.

Reviewer #2 (Remarks to the Author):

In this paper, researchers firstly built a Red2Flp2 system that is similar to the Red2Onco system which they published in 2021. Then, based on the SCON technology published in 2022, they constructed Sox2scon-FRT/scon-FRT mice line, by which this study revealed that Sox2 is not essential for adult esophagus stem cell maintenance itself, but rather for stem cell proliferation and differentiation.

This study follows the previously published Red2Onco system in 2021 and scon-loxp technology in 2022. The scon-FRT/scon-FRT mice lines and the Sox2scon-FRT/scon-FRT mice line had been mentioned in their 2022 article. This research team took full advantage of the recombinant randomness of the Confetti mice line, combined with Flp-frt for gene expression blocking, and finally performed mosaic analysis of mutant and wt cells in vivo by two recombinant enzymes combinations.

Here are some questions or concerns on the manuscript:

(1) Compared with the original Confetti mice line, does the Red2Flp show any influence on the expression of fluorescent proteins or lead to the preference of fluorescent proteins' expression? (as the RFP's signal was not high-quality in the whole paper, and the YFP signal in 2h was also not good); Is the ratio of RFP+ cells to YFP+ cells stable in Rosa-CreERT2; Red2Flp mice line under homeostatic condition?

R: we provide here, for Reviewer's eyes only, a FACS comparison of RFP positive cells sorted from confetti or Red2Flpe lines. As the reviewer pointed out the expression level of RFP is dimmer in the Red2Flpe model, compared to that of the Confetti model. However, this difference was not a major drawback when we perform in vivo lineage tracing in multiple

tissues. We believe that in combination of clearing technology and better imaging setup, one can easily get required quality of images. We feel sorry for providing relatively poor images as the reviewer pointed out.

For the RFP+:YFP+ ratio, we performed quantification of RFP+ and YFP+ cells in various tissues (see the new Extended Data Fig 2, with quantification of RFP/YFP clones in multiple organs). In conclusion, as it was observed in the original Confetti system as well as our previous Red2Onco models, we are able to observe similar number of RFP+ and YFP+ clones in all the analyzed tissues. These two colors are useful duos to be compared in the most analysis, while CFP+ clones provide additional control. Like other reports, we observe much less GFP+ cells.

(2) Line122 mentions "the utility of Red2Flpe in multiple organs and tissues with recombination rates that were suitable for mosaic lineage tracing studies". It is necessary to match their representation, multiple organs should be listed including (the lungs, heart, liver, stomach, thymus, large intestine, pancreas, kidney, etc.). Besides, since the recombination rate is mentioned, should the relative statistical results be shown?

R: We thank the Reviewer for raising this important point. As we mentioned above, we have now updated Extended Data Fig. 2, showing Red2Flpe-mediated recombination in additional organs, which are of strong interest in the field of gastrointestinal biology: colon, pancreas and stomach. We also provide quantification of Red/Yellow positive clone ratios in the same tissues, showing comparable efficiencies of recombination for the two different fluorescent proteins.

(3) The efficiency of gene knock-out needs to be shown by statistical results (High beta-catenin RFP+/RFP+; Recombination efficiency in Figure 2h also needs to be shown more directly (GFP+RFP+/ RFP+ and GFP+YFP+/ YFP+).

R: We agree with the reviewer that it is important to corroborate the efficiency of recombination in our Red2Flpe system with quantification. We point out that as for the remark about Fig2h, this was already included in the original manuscript, as we show a clear, quantified correlation between RFP (Flpe) expression and GFP expression (which is dependent on Flpe recombination) by FACS analysis. We now provide new, more clear FACS data that address this point in our new Figure 2. This analysis shows that both reporter activation and recombination are completed within 5-7 days.

As for β -catenin accumulation, this would be an indirect measure of the recombination efficiency, as it depends firstly on the successful recombination of Apc. For example, to observe clear accumulation of β -catenin, the system has to achieve complete knockout of Apc, and then extra time for the activation of the Wnt/ β -catenin pathway. This will make the system look even slower than our reporter-based analysis for recombination efficiency. In fact, we observed more heterogeneity in accumulation of β -catenin in earlier time point of induction. Nevertheless, for both Apc and Sox2 knockouts, we observed clear correlation of β -catenin accumulation and lack of Sox2 staining in all our analysis when sufficient time after induction is given. For example, we provide in below a statistical analysis of Sox2 positivity in yellow vs. red clones (Fig 5e).

if we want to use the system reported in this paper, we need to use A-CreER+/Ki; Red2Flpe+/Ki; X sc^{on}/sc^{on} mice line. The efficiency of experiments mostly depends on two key points, one is the correspondence of RFP+ expression in Red2Flpe mice line with Flp actual working efficiency; the other one is whether X sc^{on}/ sc^{on} can fully stop gene expression in vivo. None of these is shown by data in the article.

R: We thank the reviewer for raising this point. We now provide in Fig 5e a statistical analysis of Sox2 positivity in yellow vs red clones, showing recombination efficiency in a RosaCreER; Red2Flpe; Sox2^{SCON/SCON} line. We also like to emphasize that SC^{ON} knockout efficiency is expected to be more consistence among different SC^{ON} lines as they all use same SC^{ON} seq

with exactly the same distance between FRT sites, whereas the distance of FRT (or LoxP) sites influence heavily on the knockout efficiency in the conventional conditional method.

(4) Why not show the staining and statistical results (GFP+Mcherry+/ Mcherry+) of the Figure 3d results? it can be more intuitive to understand the recombination efficiency of Flp to eG-SCON-FRT-FP.

R: we apologize for not providing this quantification earlier. We now provide Violin plots in our new Fig. 3, panel d.

(5) Red2Flpe has a very high recombination efficiency for SCON-Frt in vitro (as shown in Figure 4d, 4c). But how about the efficiency of ROSA-CreERT2; Red2Flpe; eG-SCON-Frt-FP in vivo? (GFP+RFP+/RFP+)

R: We respectfully disagree to perform this particular experiment as it will need additional mouse generation and extensive crossing for getting just another confirmation of the same message. The eG-SCON-Frt-FP was used as a proof of concept in vitro, a quick but elegant way to show the effectiveness of our system. In vivo, instead, we used well established FRT-reporter system to test Red2Flpe system (Fig 2e-k). Separately, we also show the efficiency of Red2Flpe + SCON system through the quantification of Sox2 positivity in red and yellow cells (Fig 5e), which is more biological measures to show the effectiveness of our system in an endogenous gene case.

(6) It requires the zygote injection technology to build CreER+/Ki; Xscon/+ or Red2Flpe+/Ki; Xscon/+mice line. It represents a significant limitation to promoting such techniques, as similar mice lines need to be built each time.

R: Building a mosaic knockout system in mice has been a great challenge and it has never been as easy as it is presented in this manuscript. The MADM system needs chromosomal exchange and the ifgMosaic system needs construction of the cassette each time with well characterized dominant negative forms for loss-of-function studies. Comparably, our system provides well-established Red2Flpe system for labelling and red-clone specific expression of Flpe, and the SCON method provides a shortcut to the generation of FTR-based conditional alleles. This latter point was highly appreciated by the Reviewer#1.

In our hands, now in two different institutes (IMBA and IBS-CGE), we were able to generate 10-15 lines per year while setting up the SCON technology in house. We feel confident now to scale up the service to provide SCONEd mouse lines about 30-40 lines per year with our small mouse team – two zygote injectionists. The SCON technology does not require any tedious construction of DNA cassette for conditional knockout as the template (300-500 bp) for SCON insertion is fully synthesizable as well as Cas9 and sgRNA. So right after the design of targeting, all the materials can be delivered by companies. We are also working with a bioinformatics team to ease the design part of SCON technology.

We think that companies providing mouse gene editing service will easily accommodate the SCON technology in the near future as a few companies showed their interest in using our method already. We hope that all the work can be as easy as DNA subcloning and ordering antibodies for all the researchers interested in our methodology. Due to the scalable nature of the SCON technology, this renovative change is not a dream anymore.

(7) In the last part, the single-cell sequencing or bulk-seq of mut RFP+ cells and wt YFP+ cells can figure out the genetic differences and support "the esophageal cells that were depleted for Sox2 still retained their stem cell characteristics but exhibited reduced capacity for proliferation."

R: We now provide a small scale sort-seq-based analysis to show that RFP+ cells still retain a comparable or higher level of known esophageal stem cell markers (K14, p63) (Extended Data Fig 6d). We used this method due to the small number of isolated RFP+ and YFP+ cells. Moreover, this method can still provide comparable or even better depth of single cell transcriptome. As speculated, the RFP+ cells retain their stem cell characteristics.

(8) Why Sox2 exhibited a reduced capacity for esophagus stem cell proliferation but did not appear to significantly alter stem cell maintenance or clonal fitness in the stomach epithelium?

R: Genetic knockout studies show which genes are essentially needed in which context. Sometimes, the same gene can show different phenotype in different tissue for various reasons. One simple explanation is genetic redundancy among paralogues and their expression patterns. We guess that in the stomach, Sox2 might have been well compensated by another Sox protein. Nevertheless, our observation has been backed up by another report from Hochedlinger group, showing that Sox2 knockout in the stomach epithelium causes little effect but block of enterochromaffin cell differentiation (PMID: 36717627).

Sox2 was believed to be an important transcriptional factor for esophageal stem cells based on their expression pattern (PMID: 21982232) and deleterious effect of Sox2 knockout in the esophageal epithelium (PMID: 17522155). However, when mosaically deleted in our current study, Sox2 mutants showed lack of clonal expansion but sustained maintenance of stem cell program even after few weeks from Sox2 knockout as shown by immunostaining (Fig 5d, g) and scRNA-seq (Extended Data Fig 6d). This shows that Sox2 is not directly required for stem cell program.

(9) Why are the n values of YFP and RFP so different in Figure 5e?

R: We have intentionally show this by analyzing YFP and RFP clones in the same region. RFP+ clones are difficult to find over time, due to the fact that YFP+ clones persist, while RFP clones are gradually lost due to lack of clonal expansion and low fitness in cell competition dynamics, as we have explained in the text (page 12, lines 276-277; page 13, lines 315-316). Please also see the new graphs in Fig5, e and f.

(10) The system built up in this paper is based on Rosa26-Confetti (Brainbow2.1) mice line. However, they just compare mut RFP and wt YFP. What is the advantage of this system with some mosaic systems using two colors? More random?

R: We chose the Confetti system because it is a widely used and well characterized model for lineage tracing. We focused on RFP and YFP comparison simply because of the way the Confetti construct is designed: the recombination events that lead to expression of either YFP or RFP are almost MUTUALLY exclusive: this means a cell expressing YFP (WT) has very little chance to eventually convert to RFP (KO) as either one of the constructs is lost in this recombination scheme. In fact, it should be noted that although Cre-ERT2 activation is transient, it can still promote multiple recombination events, in theory, leading to expression of GFP and successively to YFP or RFP to CFP, as they are inverted to each other. Thus, the comparison between RFP and YFP is the most accurate choice.

Reviewer #3 (Remarks to the Author):

In their study, Wu and colleagues have developed a versatile set of tools to efficiently generate mouse models for mosaic knockouts within a short period of time. The purpose of these models is to simultaneously trace the lineage and study the fate of mutant and wild-type clones. Previously, they reported the development of the Red2Onco mouse model (Yum et al., Nature 2021), which is a confetti/brainbow-like model. In this model, cre

activity stochastically induces the expression of CFP, GFP, YFP, or RFP, and the latter one along with the co-expression of an oncogene. This model enables the mosaic induction of RFP mutant clones and wild-type CFP, GFP, and YFP clones. In their current study, the authors introduce a new model called Red2Flpe, where they have replaced the oncogene with Flpe. This novel model allows for the Cre-mediated induction of Flpe activity in the RFP clones. Importantly, by combining this model with their recently developed SCON models (models in which Flpe activity induces the conditional loss of a gene; Wu et al., Exp Mol Med 2022), the Red2Flpe model enables the induction of RFP expression and concurrent loss of a (tumor suppressor) gene. This represents a significant advancement for the field as it allows for the investigation of the fate and behavior of mutant cells lacking (tumor suppressor) genes, enabling comparisons with wild-type cells.

To illustrate the great potential of their new model, the authors successfully labeled Apc mutant cells in the intestine and correlated them with β -catenin accumulation. Additionally, they demonstrated that Sox2 is not an essential factor for stem cell maintenance, but it plays a crucial role in the proliferation and differentiation capacity of the gastric epithelium.

This paper describes an impressive improvement in our current toolkit, with technically well-executed experiments and a remarkable tour-de-force. Moreover, the authors reveal interesting new insights into the role of Sox2 in stem cell potential in the esophagus and stomach. Overall, I highly recommend publishing this manuscript in Nature Communications, with the condition that the authors address the following minor points:

R: We appreciate the Reviewer's comments and we are happy for his/her support our newly described genetic tool.

Minor Comments:

Extended Data Fig. 2: The authors demonstrate the utility of Red2Flpe in seminal vesicles, spleen, and tongue. However, it would be valuable to include other more commonly studied organs such as the skin and intestine.

R: We thank the Reviewer for raising this important point. We have now updated Extended Data Fig. 2, showing Red2Flpe-mediated recombination in additional organs, which are of strong interest in the field of gastrointestinal biology: colon, pancreas and stomach. We also provide quantification of Red/Yellow positive clone ratios in the same tissues, showing comparable efficiencies of recombination for the two different fluorescent proteins.

In Figures 2b, 2c, and 2d: The authors convincingly demonstrate β -catenin stabilization in RFP+ cells after a week of Tamoxifen-mediated Red2Flpe induction. It would be beneficial for the field to explore the timing of β -catenin accumulation. Is there a correlation between the number of mutant cells and β -catenin upregulation? How does β -catenin stabilization evolve over time in this mouse model? In other words, could the authors provide further insights into the heterogeneity of β -catenin accumulation in RFP+ cells from day 5 to day 10?

R: While we agree with the Reviewer that this could be an interesting analysis, we argue that it is technically challenging to reliably quantify β -catenin accumulation in vivo. Indeed, we do see heterogeneity of β -catenin among RFP+ clones, but this heterogeneity is likely caused by stochastic timing of Flp-mediated recombinations or by some other biological effects, including cell competition between mutant and WT clones. Due to this uncertainty, we would rather avoid making a strong point on this. An important point here is that for both Apc and Sox2, we see clear correlation between RFP expression and β -catenin accumulation and loss of Sox2 positivity, demonstrating the utility of Red2Flp system in WT/MT labeling and lineage tracing.

Fig. 3D: Does the SCON-FRT system lead to complete loss of GFP or a significant reduction? If it results in complete loss, could the authors include a plot showing an experiment where cells did not express any construct (negative control)? If there is partial loss of GFP, it would be helpful to report on this. In many

experiments, partial loss may even be favorable, and it is important to know whether the SCON-FRT system leads to knockdown or knockout.

R: We can confirm that it is a complete loss of GFP fluorescence. As suggested by the reviewer, we have included an empty vector negative control in the updated Figure 3, panel c, and we added violin plots (panel d) displaying the quantification of GFP intensity with or without Flp recombinase.

In Figures 2i, 2j, and 2k: The authors present a time series of flow cytometry to isolate GFP/RFP cells and demonstrate the long maturation time of the Tdimer2 fluorophore. However, in Figure 2h, not every RFP cell is GFP positive. Could it be possible that the gating strategy of their flow cytometry setup is too stringent? It would be helpful to include images at various time points (e.g., day 5, 7, and 10) and analyze the GFP/RFP ratio in each image.

R: Indeed, we used a stringent gating strategy. This was necessary, as during the analysis we observed noise that could be excluded only with this stringent setup. It is true that not every RFP+ cell was also GFP+, but this was only a relatively small fraction. We now provide an updated Figure 2, where we show that at Day 7 and 10 after recombination, the ratio of double positive RFP+/GFP+ cells is above 80%. Overall, we show that the efficiency of Flpe-mediated recombination is high, as we provide examples in APC mosaic clones, which show increased β -catenin levels in RFP+ clones, as well as Sox2 KO (updated Figure 5) where all the RFP+ cells lost Sox2 positivity (Fig 5e) with clear phenotype.

Figure 5: In this section, the authors investigate the role of Sox2 in stem cell maintenance. Interestingly, they demonstrate that Sox2 is not crucial for stem cell maintenance, as RFP+ clones remained in the tissue even after long-term tracing. However, is the loss of RFP+ clones not dependent on the turnover rate of cells in the esophagus? Slow turnover may potentially result in a gradual loss of clones. Furthermore, the authors find that Sox2 mutant cells have a lower organoid-forming efficiency compared to wild-type cells. Organoid-forming capacity is often considered an indicator of stem cell potential in the field. Therefore, it might be advisable to temper the conclusion slightly regarding the lack of effect of Sox2 knockout on stem cell maintenance.

R: Indeed, as predicted by the reviewer we saw greater loss of RFP+ clones, although at early time points (2.5 weeks) the number of RFP (Sox2 KO) and YFP (WT) clones are comparable (see new Figure 5). This is due to the lack of proliferation and clonal expansion, which leads to lower fitness and causes depletion of Sox2 KO RFP+ clones from the tissue (Fig 5f). In this figure, YFP+ clones are also being lost through the replacement of non-labelled WT cells. Sox2 KO RFP+ cells suffer more from this competition and show faster depletion from the tissue. However, as we speculated the expression of esophageal stem cell markers (K14, p63) was not changed in the Sox2 KO RFP+ cells when analyzed in our scRNA-seq data. It means that Sox2 mainly influenced the clonal expansion capacity rather than the esophageal stem cell program as we initially proposed. Nevertheless, we agree with the reviewer in tempering our conclusion accordingly.

REVIEWERS' COMMENTS

Reviewer #2 (Remarks to the Author):

Authors have addressed my questions adequately during the revision.

Reviewer #3 (Remarks to the Author):

The authors have done an excellent job addressing all my minor concerns. As I mentioned previously, this paper describes an impressive improvement in our current toolkit, with technically well-executed experiments and a remarkable tour de force. I strongly recommend publishing this exciting paper in Nature Communications.